# Patients' and nurses' views on providing psychological support within cardiac rehabilitation programmes: a qualitative study

Katrina M Turner,[1,2] Rachel Winder,[3] John L Campbell,[3] David A Richards,[3] Manish Gandhi,[4] Chris M Dickens,[3] Suzanne Richards[3,5]

[1]Population Health Sciences, University of Bristol, Bristol, UK
[2]The National Institute for Health Research Collaboration for Leadership in Applied Health Research and Care West (NIHR CLAHRC West), Bristol, UK
[3]University of Exeter Medical School, Exeter, Devon, UK
[4]Royal Devon and Exeter NHS Foundation Trust, Exeter, UK
[5]Leeds Institute of Health Research, Leeds, UK

**Correspondence to**
Dr Katrina M Turner;
Katrina.Turner@bristol.ac.uk

## ABSTRACT

**Objective** To explore patients' and nurses' views on the feasibility and acceptability of providing psychological care within cardiac rehabilitation services.

**Design** In-depth interviews analysed thematically.

**Participants** 18 patients and 7 cardiac nurses taking part in a pilot trial (CADENCE) of an enhanced psychological care intervention delivered within cardiac rehabilitation programmes by nurses to patients with symptoms of depression.

**Setting** Cardiac services based in the South West of England and the East Midlands, UK.

**Results** Patients and nurses viewed psychological support as central to good cardiac rehabilitation. Patients' accounts highlighted the significant and immediate adverse effect a cardiac event can have on an individual's mental well-being. They also showed that patients valued nurses attending to both their mental and physical health, and felt this was essential to their overall recovery. Nurses were committed to providing psychological support, believed it benefited patients, and advocated for this support to be delivered within cardiac rehabilitation programmes rather than within a parallel healthcare service. However, nurses were time-constrained and found it challenging to provide psychological care within their existing workloads.

**Conclusions** Both patients and nurses highly value psychological support being delivered within cardiac rehabilitation programmes but resource constraints raise barriers to implementation. Consideration, therefore, should be given to alternative forms of delivery which do not rely solely on nurses to enable patients to receive psychological support during cardiac rehabilitation.

**Trial registration number** ISCTRN34701576.

## Strengths and limitations of this study

► This is the first study to detail both patients' and nurses' views on providing psychological care within cardiac rehabilitation services.
► Interviews were held with patients receiving care in seven different cardiac rehabilitation teams, and with nurses from four of these teams.
► Employing in-depth interviews ensured interviewees could raise issues that were salient to them and not predicted by the research team.
► The relatively small number of patients and nurses recruited to the CADENCE pilot trial limited the number of interviews that could be conducted, and the extent to which individuals could be purposefully sample to ensure maximum variation within the sample.
► The views expressed by interviewees about the provision of psychological care within cardiac rehabilitation services may have been influenced by their experiences of the CADENCE trial.

include the association between depression and cardiac risk factors (eg, hypertension, smoking and reduced physical activity), greater coronary disease severity, treatment non-adherence to cardiac medication and rehabilitation programmes, and increased platelet aggregation.[8 9] There is national and international recognition that the detection and treatment of depression among these patients is important.[10–12]

Routine clinical care for patients with CHD who have experienced acute coronary syndrome includes the provision of cardiac rehabilitation. The British Association for Cardiovascular Prevention and Rehabilitation's guidance[10] states that usual cardiac rehabilitation should include psychological support. The majority of people attending cardiac rehabilitation in the UK, however, do not receive psychological care.[13]

## BACKGROUND

About 20% of individuals with coronary heart disease (CHD) report symptoms of depression.[1] This proportion is approximately four times greater than the levels identified within the general population.[2] Depression among patients with CHD is associated with greater risk of subsequent cardiac morbidity and mortality.[3–7] Reasons for this association remain unclear but possible mechanisms

Cardiac rehabilitation programmes usually involve an initial assessment followed by a structured programme that lasts between 6 and 8 weeks. This programme may include clinic appointments where patients' cardiac symptoms are monitored and discussed, supervised exercise sessions and educational talks. Programmes are delivered primarily by cardiac nurse specialists, who are supported by physiotherapists. In terms of how nurses can best provide psychological support to patients undergoing rehabilitation, possible models include them delivering psychological support within the structured programme, referring patients onto other mainstream health services providing treatment for depression, and/or external mental health practitioners working closely with cardiac nurses to deliver psychological care to their patients. It is not known which approach would be most acceptable to patients and nurses.

Little is known about patients' and nurses' views and experiences of receiving/delivering psychological support within cardiac rehabilitation programmes. Simmonds et al[14] explored patients' views and experiences of living with depression and CHD, and Pâquet et al[15] assessed patients' experiences of care received during the first 3 months following a cardiac event requiring hospitalisation, but neither study explored patients' views of any formal psychological support they had received during their rehabilitation, nor how they thought this care should be given. To date, no study has documented cardiac nurses' views of this area.

The recently completed CADENCE study developed and piloted an enhanced psychological care (EPC) intervention for patients presenting with depressive symptoms following a cardiac event who attended cardiac rehabilitation.[16] EPC was delivered by cardiac nurse specialists, within their existing workloads and embedded within the structured cardiac rehabilitation programme. It consisted of mental healthcare coordination and a patient-led, nurse-supported programme of behavioural activation (BA). The care co-ordination aspect of the intervention was based on current National Institute for Health and Care Excellence (NICE) guidance.[17 18] BA is a simple psychological treatment for depression that aims to re-engage patients with positively reinforcing experiences and reduce avoidance behaviours.[19] It is no less clinically effective but more cost effective than cognitive behavioural therapy in treating depression in adults.[20]

The CADENCE study included a feasibility study and external pilot cluster randomised controlled trial. As part of the pilot trial, in-depth interviews were held with patients and nurses who, as part of the study, had been trained to deliver EPC. They explored patients' and nurses' views on the provision of psychological support within cardiac rehabilitation programmes and, where appropriate, their experiences of receiving/implementing EPC. This paper details findings from these interviews. It reports patients' and nurses' views on the feasibility and acceptability of providing psychological care within cardiac rehabilitation

services, reflecting on how such care could be most appropriately delivered in the future.

## METHODS
### The CADENCE pilot trial
Twenty cardiac rehabilitation teams were approached to take part in the pilot trial between December 2014 and February 2015. Eight teams agreed to participate. They were based in the South West of England or the East Midlands. Five of the participating teams were randomised to EPC and three to usual care (UC). Nurses in all eight participating teams were responsible for screening and recruiting patients.

Patients were eligible to take part if they were ≥18 years old, referred for cardiac rehabilitation based on local clinical referral protocols and scored ≥10 on the Patient Health Questionnaire-9 (PHQ-9).[21] Patients were not eligible if they reported being treated for depression in the 6 months before their acute cardiac event; if there was evidence of alcohol or drug dependency; where the participant was acutely suicidal; or where there was evidence of poorly controlled bipolar disorder or psychosis/psychotic symptoms.

Eleven cardiac rehabilitation nurses working in the five intervention teams were trained, over 2 days, in how to deliver EPC. The training covered mental healthcare coordination, how to support patients working with a BA self-help manual, assessment and management of psychiatric risk, and how to use the CADENCE materials, for example, the clinical materials to be used during BA sessions. While delivering EPC to patients, each nurse received clinical supervision every fortnight from an accredited BA therapist. This was held by telephone, either on an individual basis or in conjunction with other study nurses within their own team.

In order to assess treatment fidelity, nurses were asked to complete a structured form after delivering EPC to patients. This form had been developed by the research team and invited nurses to record which elements of the EPC intervention they had delivered. These notes were reviewed by members of the research team, alongside feedback from the clinical supervision sessions. Based on this information, the research team concluded that the nurses had delivered EPC as intended.

Seven of the participating teams managed to recruit patients to the trial. In total, 29 patients were recruited (15 EPC and 14 UC). All patients and nurses involved in the trial, at the time of being recruited to the study, agreed to their contact details being passed onto the qualitative research team so they could be approached for interview.

### Patient and nurse interviews
Patients were interviewed once they had completed their 5 month follow-up in the trial, in case the interview influenced their views of the intervention or the study. Also, by this time, most individuals had completed their cardiac rehabilitation. The intention was to purposefully sample

interviewees on the basis of age, gender, recruiting team and, in the case of the EPC participants, adherence to EPC. However, as it took longer than expected to recruit patients to the trial, in the end, all 15 EPC patients and 6 of the UC patients were approached for interview.

Two of the EPC patients approached were uncontactable and another EPC patient declined to be interviewed. Thus, interviews were held with a total of 18 patients (12 EPC and 6 UC). These interviewees were recruited across the seven participating teams who had recruited patients to the trial. Patients were interviewed by RW, a researcher with over 10 years' experience of conducting qualitative research, between January and June 2016. The initial two interviews were held on a face-to-face basis. The remaining interviews were held by telephone, as interviewees were based across a large geographical area and because well-planned telephone interviews can gather the same material as interviews held face-to-face.[22] The two individuals interviewed in person provided written consent to take part at the time of interview. Individuals interviewed over the telephone were posted a consent form and asked to complete and return it to RW, using a prepaid envelope, prior to interview.

A topic guide was used to ensure consistency across the interviews. It was based on the aims of the research, a review of relevant literature, and the researchers' knowledge of cardiac rehabilitation and the intervention. Two versions of the guide were used: one for the EPC arm and one for the UC arm. Both included questions about the patient's cardiac event, experience of cardiac rehabilitation, mental well-being during treatment and relationship with their cardiac nurse. The version used with the EPC arm also included questions about experience of EPC and treatment adherence.

Eight of the 18 patients interviewed were male and all the interviewees reported their ethnicity as White British. The mean age of interviewees was 67.0 (age range 50–79 years) and their average pretreatment PHQ-9 score was 13.9. Cardiac events reported by the patients included heart attacks, angina, cardiomyopathy, heart failure and valve disorder treated through coronary artery bypass grafts, stent insertion, valve replacement, medication or a combination of these. All interviewees had started their cardiac rehabilitation programme and attended at least one cardiac nurse appointment. The interviews with EPC patients on average lasted longer than those held with UC patients (52 vs 32 minutes).

Seven nurses (all females) from four of the five intervention teams were interviewed. No nurses from the fifth team were interviewed as this team did not recruit any patients. Nurses were interviewed once they had delivered EPC to one or more patients. A topic guide was used and covered the following areas: experience of delivering EPC, dealing with patients at risk of suicide or self-harm, and impact of delivering psychological care on their relationship with patients.

The nurse interviews were held by RW between December 2015 and March 2016. RW was known to the nurses, as she had observed the two training days. The first three nurse interviews were conducted in person. The remaining four were held by telephone, as these nurses were geographically dispersed. As in the case of the patient interviews, nurses provided a written consent to be interviewed at the time of interview, or completed and returned a consent form prior to interview, depending on whether they were interviewed in person or over the telephone. At the time of interview, each nurse had delivered EPC to between one and four patients. The interviews lasted between 38 and 64 minutes.

Ethical approval to conduct the pilot trial and the qualitative work nested within it was given by NRES Committee South West, Exeter (reference: 14/SW/0139).

### Data analysis

Data collection and analysis proceeded in parallel, so that early insights could inform the focus of later interviews. Both patient and nurse interviews were audio recorded and fully transcribed. Both interview sets were analysed thematically, as this allowed comparisons to be made within and across the interviews and highlighted patients' and nurses' views towards specific issues, for example, the feasibility of cardiac nurses providing psychological care.

Initially, KMT and RW independently read a sample of nurse and patient transcripts in order to identify emerging themes and to develop a preliminary coding frame. They then met to discuss their coding. This discussion led to two coding frames being drafted: one for each dataset. Where possible, similar codes were used within each coding frame to assist triangulation of nurses' and patients' accounts. Once the coding frames had been agreed, transcripts were manually coded and data pertaining to each code summarised in tables using an approach based on framework analysis.[23] The researchers then read and re-read the tables to identify key themes and deviant cases, and to highlight similarities and differences between the datasets.

## RESULTS

Analysis of the data highlighted patients' and nurses' views on the importance and benefits of providing psychological care within cardiac rehabilitation programmes, indicated the extent to which patients and nurses felt it was acceptable and practical for nurses to provide this care, and identified issues relevant to service planners regarding when, where and how psychological support should be provided in cardiac rehabilitation.

Below, quotes have been reproduced to illustrate key points. They have been tagged according to whether a patient or nurse is being quoted, and using the interviewee's assigned identification (ID) number.

### The need for psychological support

Both EPC and UC patients described the significant impact their cardiac event had had on their emotional well-being. Most reported experiencing a dramatic loss of

self-confidence, panic attacks, sleeplessness and/or a lack of energy and motivation. One individual stated that he had felt *'mentally and physically shell-shocked'* (EPC, patient 12) and others described how they had been suicidal:

*'It just all came out and I was surprised at myself, to be honest, I had no idea that I was, my overwhelming feeling was disappointment that I'd actually survived and that frightened me because I've never felt that before [suicidal]…'* EPC, patient 14

*'I don't show it [how they feel]… you feel really drained and really start thinking the worst and what's the easiest way out which is, and then you think 'which is the easiest way out tablets or…'[speech trailed off].'* EPC, patient 9

The intensity of this emotional response may have been partly due to the fact that patients were aware of the contrast between their former and current self, and what they had lost following their cardiac event:

*'It was a horrible, horrible experience, I just felt my life was ebbing away, and I've never felt that low before… I had no go in me, no energy, no focus, horrible, horrible sensation.'* UC, patient 2

*'I was angry, frightened, upset… I got very depressed, I lost my job, I loved my job… suddenly everything had been ripped out from under my feet and I got very depressed, very anxious and felt a failure.'* EPC, patient 17

Some patients described feeling supported by family and friends, while others detailed how they had been unable to discuss how they felt with others. In addition, most patients reported the time between being discharged from hospital to their first cardiac rehabilitation appointment as particularly difficult. Several patients described feeling lonely as they had very little or no family around, and others recalled feeling unable to cope after the '*cosseted*' (EPC patient 13) hospital environment and cut '*adrift*' (EPC patient 14) from professional help on hospital discharge.

When focusing on the nurses' accounts, it was apparent that all the nurses viewed psychological care as a core component of cardiac rehabilitation and had provided psychological support prior to their involvement in CADENCE. Nurses had routinely screened for depression and anxiety using Hospital Anxiety and Depression Scale (HADS).[24] In terms of the content of psychological support provided, there was local variation. Most nurses had provided support by giving talks on stress and relaxation, and some had also referred patients to their general practitioner (GP) or encouraged patients to self-refer to Improving Access to Psychological Therapies (IAPT) services, which some nurses described as having good relations with. Interestingly, one nurse's comments implied the psychological aspects of cardiac rehabilitation could dominate over the physical:

*'I've got to be honest, I mean, sometimes I've left a cardiac rehabilitation clinic and all that we have addressed is the psychological side of things.'* Nurse 5

Having been asked within the trial to screen patients using the PHQ-9, which includes a question specifically about suicidal thoughts, nurses became aware that when previously using the HADS, they might not have identified patients who were at risk:

*'Now it's a lot more formalised and I'm more aware of that patient, because before I wasn't picking them up, because I wasn't doing the PHQ-9.'* Nurse 1

Some nurses described being surprised by how frequently they now identified patients with such thoughts, and how completing the questionnaire provided patients with an opportunity to discuss their mental health which would often *'come a bit out of the blue'* (Nurse 4).

### The acceptability and feasibility of nurses providing psychological care

Both EPC and UC patients said that they had developed good relationships with their cardiac nurse and viewed them as someone they could talk to. The UC patients, however, described how they had not discussed their mental well-being with their nurse in any detail, if at all.

Interviewer: *'Did the nurse talk to you at all about your anxiety at the time or anything that she could help you with, did she give you a leaflet or did she talk about it?'*

UC, patient 1: *'No, no, because I don't expect she knew about it. Unless she knew about it and didn't say nothing. As far as I can remember, it was never ever talked about.'*

Three of the UC patients mentioned they would have liked to have spoken to their nurse about their mental well-being, but then implied that they had not been given the opportunity to do so.

*'They [the nurses] are doing a brilliant thing, I cannot fault them… but they're very structured, they do what they do to help you and it's a great thing… it's just when you have got anxieties or you're worried about something… it's a shame, I felt that there wasn't enough of the one to one.'* UC, patient 2

EPC patients reported receiving EPC before, during or after attending their rehabilitation fitness session. They had welcomed the opportunity to talk about their emotions, felt comfortable with their nurse providing both physical and mental support and were confident in the nurse's ability to do this. It was also apparent that they valued the focus of their care encompassing both their mental and physical health:

*'It was a good surprise in that I thought the cardiac rehab would just be 'Hello! Do a few exercises' and that's it. I thought it was really brilliant, that it was looked at holistically, brilliant. I came out thinking, "oh this is a real breath of fresh air, that people should actually look at me as a total person and not just as a patient with a dodgy heart' sort of thing."'* EPC, patient 13

*'To have somebody that deals with the mental health side as well as the physical I think is incredibly important… it*

*shouldn't be just about rehabilitating the body, it should be rehabilitating the mind as well.'* EPC, patient 17

EPC patients viewed the one-to-one dedicated time they had had with their nurse to discuss their mental health as crucial to their physical and mental recovery, and two participants who had experienced suicidal thoughts described this input as lifesaving:

*'Without people like you and [name of nurse] talking to me and me talking to you, I might not be here now.'* EPC, patient 8

*'I think a lot of people when they commit suicide and things like that… they just feel useless to everybody and nobody and the good thing with that* [EPC] *is they do help you… it's certainly helped me.'* EPC, patient 9

Talking to a nurse had allowed patients to express how they felt, to better understand why they felt low, and had encouraged them to talk to others. In addition, EPC patients described how BA had enabled them to see the link between their mood and certain activities, which in turn had led to them changing or increasing activities to improve their mood.

However, despite valuing the care they had received, some EPC patients commented that they would have liked more psychological support but were aware of the '*tremendous pressure*' (EPC patient 12) nurses were working under, so felt it would be inappropriate to ask for more time. Also, not all patients felt nurses were best placed to deal with their mental health needs. For example, one patient commented that she did not want to talk about mental health with a 'physical' health practitioner. In addition, several patients mentioned they had received their EPC session in a corner of the gym or leisure centre because the nurse did not have access to a private room. Although most participants said they were not worried about the lack of privacy, one participant described how she felt uncomfortable discussing her emotions in public.

Comments by nurses indicated that they felt capable of providing psychological care. They described the training they had received as part of CADENCE as covering everything they needed to know and stated that, with experience, they had found ways to deliver EPC smoothly. In addition, one nurse commented that patients learnt quickly and implied that as patients progressed through treatment, less work was involved:

*'Within a couple of weeks I think the patients saw very quickly how their mood was related to their activities and they got it very quickly. So after that actually the time spent with them prior to the group was more as you say coordinating them to the next phase… and actually moving them onto actually allocating routine activities.'* Nurse 2

Nurses commented that, having completed the CADENCE training, they felt more able to effectively treat depression and had formalised their approach. They also said they now spent more designated time with patients discussing mental health, which they felt patients valued,

and thought BA had encouraged patients to be more physically and socially active. However, nurses described how delivering EPC could be time consuming. An EPC session incorporating the BA component could take between 10 and 40 minutes, and even longer if the PHQ-9 indicated that the patient was 'at risk'. Time was also needed to receive clinical supervision and to chase patients who had not attended. The impact of this increased workload meant nurses shortened their lunch breaks, delayed going home, ran late with other patients and/or asked a colleague to '*double up*' (Nurse 6) so one of them could focus on the EPC patient while the other took the fitness session.

This increase in workload also meant that some nurses felt '*lucky*' (Nurses 1, 2, 6) that they had not needed to see more than one EPC patient at the same rehabilitation session or on the same day. One nurse (2) also said it would be '*mind-numbingly brain taxing*' to deliver EPC to more than one person in the same session or on the same day, suggesting nurses would also have experienced this situation as mentally and emotionally demanding.

In addition to having limited time, nurses reported having little or no access to a private room where they could talk to patients about sensitive issues. Most nurses worked across various sites, either within their hospital or within the community, for example, in leisure centres or health centres. A lack of space led to nurses asking EPC patients to come early to a rehabilitation session or to stay on afterwards, and talking to patients in a quiet corner of the fitness room. The latter situation was viewed as not ideal but perhaps less intimidating for patients than meeting in a private room.

### When and how to provide psychological care for depression

Patients felt that embedding psychological support within cardiac rehabilitation for patients with symptoms of depression was timely and appropriate. However, some commented that they would also have liked to have commenced such care earlier, either when in hospital following their cardiac event or immediately following hospital discharge, as they had experienced this time as particularly difficult. Patients also thought psychological support should be continued beyond cardiac rehabilitation if it was still needed:

*'I think additional help after this [end of EPC] is a definite must for some people, it was for me… if they [your nurse] feel that you need more help, that should be offered definitely. Or if it's a waste of time, cos you just literally will go back to how you were before, I think.'* EPC, patient 11

Some patients suggested that some aspects of EPC could be delivered within a group, as they had found it reassuring to talk to other patients about their experiences and to hear about theirs. However, other patients thought a group environment might inhibit what individuals discussed, acknowledging that they themselves would struggle to publicaly talk about their emotions.

Some patients had been care coordinated to other services, for example, to their GP or to IAPT services,

and had found this beneficial. However, six EPC patients said they would not discuss their mental health with their GP, believing that their GP would only prescribe an antidepressant:

*'I'd say what's the point in me going to the doctors, he gives me these bloody tablets, I'm not going to live on sodding tablets, what's the point?'* EPC, patient 9

Nurses commented that patients who had scored just <10 on the PHQ-9 may have benefited from receiving psychological support. They also raised the possibility of delivering psychological support to groups rather than individual patients, as they too realised that individuals could benefit from talking about their experiences with others:

*'Patients get a huge amount of benefit just in talking to each other, don't they, and so the problem, the trouble solving, the solutions, "oh I do this," and just seeing how other people are getting on, the little supportive networks that they strike up when they're actually in the waiting room waiting for us to assess them and they've already got their own counselling and social network going on there, so I do recognise the power of actually getting them together as a group.'* Nurse 5

Yet, like patients, they also acknowledged that some patients would not want to discuss their personal views in front of others. A suggested possible solution was to have an introductory EPC session in a group setting and then offer one-to-one sessions to explore personal issues.

In terms of where to provide psychological support, nurses were aware that often patients did not want to be referred to their GP and described how patients had declined referrals to other services for their mental health. They also felt that integrating psychological care within cardiac rehabilitation had led to patients being more receptive and willing to address their mental health:

*'…ninety percent of the time people would say "no I don't want to do that [be referred] I'm just going to work through it, I'm going to see how I get on." So actually being able to offer an extra option that didn't involve all of that, people were more receptive.'* Nurse 2

Nurses also remarked that when they had referred patients to another service, they were not informed whether or not the patients had attended. Thus, although recognised as a challenge, most nurses commented that if additional resources could be made available, then integrating psychological support, such as EPC, within cardiac rehabilitation programmes would be the ideal situation.

## DISCUSSION

Both patients' and nurses' accounts highlighted the need to provide psychological support to patients with symptoms of depression receiving cardiac rehabilitation. Patients detailed the significant emotional impact their cardiac event had on them, and described psychological support as not only key to their mental recovery but also

as supporting their physical well-being. Prior to their involvement in CADENCE, nurses had viewed psychological care as an essential part of their role. The training they had received during the CADENCE trial had led them to formalise their approach and to spend more dedicated one-to-one time with patients.

Both patients' and nurses' accounts suggested that psychological support should be embedded within cardiac rehabilitation programmes, rather than provided out with through a parallel service, and that EPC appeared effective in treating symptoms of depression in this patient group. However, it was apparent that nurses were time-constrained and found it challenging to provide psychological support within their existing workloads.

In terms of future delivery of psychological support, this study raised issues relevant to when, where and how this could be done. Patients interviewed argued for psychological support to commence earlier, either during hospital immediately following a cardiac event or on hospital discharge. On average, most patients spend only 3 days in hospital following a heart attack[25] and are discharged back to the community with no formal support during the initial weeks prior to commencing cardiac rehabilitation. Patients in this study talked about feeling lonely and cut adrift from professional help following hospital discharge, and this appeared to compound their low mood. They also described a sense of loss: loss of ability, loss of confidence and a loss of roles. The theme of loss was one that underpinned all the interviews conducted by Simmonds *et al*[14] in their study of patients' views and experiences of living with depression and CHD. It is also discussed by Barley *et al*,[26] who report primary care practitioners viewing patients' loss of a valued role or ability to fulfil responsibilities as contributing to the development of depression following a cardiac event. In addition to suggesting that psychological support commenced earlier, patients also suggested it to be continued after discharge from a rehabilitation programme where necessary. Although evidence suggests depression after cardiac rehabilitation is not common, when present it is usually associated with other forms of psychological stress.[27] As there is an association between psychological stress and post-cardiac rehabilitation morality, it has been suggested that patients with CHD are assessed for psychological risk factors both prior to and after receiving cardiac rehabilitation.[27]

Our findings suggest psychological support should be embedded within cardiac rehabilitation programmes, as patients welcomed cardiac nurses attending to both their physical and mental well-being, viewing this as providing a more holistic approach. In addition, nurses felt integrating psychological support within existing programmes encouraged patients to acknowledge their need for psychological input. Nurses were also aware that patients often declined referrals to other services, and both patients and nurses mentioned patients' reluctance to consult a GP about their mental health.

Whole system approaches that integrate mental and physical healthcare are viewed as the most appropriate way to support patients living with mental and physical morbidity,[28] and other researchers have reported patients with CHD and depression as being ambivalent about seeking help from a GP.[14] Here patients linked this reluctance to their assumption that a GP would prescribe an antidepressant. Yet GPs may be hesitant to prescribe antidepressants for patients with CHD, as they are aware that patients may be unwilling to take an antidepressant and view other forms of treatment that encourage patients to be physically and socially active as potentially more effective for depression in this patient population, for example, exercise on referral and cardiac rehabilitation.[26]

Another reason patients might be reluctant to seek help is the stigma that surrounds mental health. Nurses reported patients only talking about how they felt once the nurse had screened them as positive for symptoms of depression and/or at risk of suicide or self-harm. Non-disclosure of mental health problems by patients with physical conditions can compound management problems.[29] Here it was apparent that nurses had been able to identify patients at risk because they had been asked to screen patients using the PHQ-9, which explicitly asks about suicidal ideation. Thus, cardiac teams may want to consider using the PHQ-9 in the future, alongside training to ensure nurses manage self-harm risk appropriately and in a way that is consistent with clinical guidelines.[17 18]

Although both patients and nurses advocated for psychological support to be provided within rehabilitation programmes, both acknowledged that nurses found it challenging to provide this care within their existing workloads. Group-based approaches were suggested, as both sets of interviewees were aware that patients could benefit from interacting with other cardiac patients. This could reduce nurse workload but whether group delivery would be possible would partly depend on the intervention. For an intervention such as EPC, where there were both general and patient-specific components, it might be possible for nurses to deliver some aspects of the intervention in a group setting and other components on a one-to-one. For example, BA could be discussed during group educational talks and care-coordination delivered on an individual basis. Doing so would ensure all patients receiving cardiac rehabilitation received information on BA; a move which would address the view expressed by nurses here that psychological support could benefit all patients and not just those who had screened positive for depression. Such an approach would concur with the recent findings of Blumenthal et al,[30] who observed the potential benefit to longer term psychological morbidity when stress management was offered to all patients following an acute cardiac event.

In terms of what psychological support should be provided, this paper did not focus on patients' and nurses' specific views of EPC, as the aim was to assess their more general views on provision of psychological support. However, it was evident that nurses and patients viewed the patient-led, nurse-supported BA component of the intervention as effective in helping patients manage their depression, and this component fits with NICE guidance for treatment of depression in people with physical health problems, as it proposes that individuals with depression and physical health problems start on low-intensity treatments, including guided self-help and physical activation.[18] In addition, a recent process evaluation reports that patients who have received BA for depression perceive it as leading to both cognitive and behavioural changes, which they view as improving their symptoms and also their lives more broadly.[31] In terms of care coordination, while our findings suggest psychological care should be embedded within cardiac rehabilitation programmes, some patients had benefited from being referred to other services and this did give patients greater treatment choice. Lastly, the intervention was developed and revised in response to comments from patients and nurses involved in the feasibility study that preceded the CADENCE pilot trial.[32] Thus, EPC is more likely than other interventions to be acceptable to patients and nurses, although findings from the pilot trial showed that it remained too burdensome for nurses to deliver long term.

### Study's strengths and weaknesses

Individuals recruited to the CADENCE pilot trial needed to score ≥10 on the PHQ-9 in order to be eligible to take part. Strictly speaking this score reflects depressive symptoms, rather than a formal clinical diagnosis of depression. However, in the UK, front-line primary care mental health services (eg, GPs and IAPT mental health workers) routinely use the PHQ-9 to diagnose and actively manage/treat depression. In addition, the PHQ-9 has been found to be as good as a diagnostic gold standard in detecting depression.[33]

The relatively small number of patients and nurses recruited to the trial limited who could be approached for interview and thus the possibility of sampling individuals purposefully to ensure maximum variation within the sample in relation to participant characteristics. This means we cannot be confident that data saturation was reached. It also means certain groups of individuals are not represented, for example, none of the patients interviewed were from an ethic minority. This might limit generalisability of the findings to the wider population of patients or nurses using cardiac rehabilitation services in the UK. However, we interviewed patients from seven different teams and nurses from four of the five participating intervention teams. As many of the key findings were evident across the interviews, there is little reason to think that the findings reported here would not be relevant to other cardiac rehabilitation programmes.

### Conclusions and implications

Both patients and nurses highly value psychological support being delivered by nurses within cardiac

rehabilitation programmes, but time and resource constraints raise significant barriers in terms of implementation, so alternative approaches need to be considered. Given that nurses viewed provision within rehabilitation programmes as ideal and mentioned good relationships with local IAPT services, an alternative approach could be nurses co-ordinating IAPT-trained psychological well-being practitioners to provide psychological support within cardiac programmes. In terms of treatment offered, it could include BA, as this was viewed as potentially effective by patients and nurses and is a treatment that can be successfully delivered by junior mental health workers.[20]

**Acknowledgements** The authors would like to thank all the nurses and patients who agreed to be interviewed as part of their involvement in the CADENCE pilot trial. They would also like to acknowledge members of the CADENCE team who are not authors on this paper but were involved in the CADENCE pilot trial and supported the qualitative work: Professor Rob Anderson, Mrs Antoinette Davey, Professor Andrew Gibson, Dr David Kessler, Mr Luke Knight, Professor Willem Kuyken, Professor Rod Taylor, Dr Obioha Ukoumunne, Dr Fiona Warren, Ms Julie Chudley and Dr Christine Wright.

**Contributors** KMT is the principal investigator for the qualitative work nested within the CADENCE study and led on writing this manuscript. RW conducted and analysed the interviews on which the paper is based. JLC is the chief investigator for the CADENCE study. SR is the scientific lead. DAR, CMD and MG were the other members of the CADENCE team. All authors contributed to editing of the final manuscript and the refining of its intellectual content.

**Funding** This work was funded by the UK NIHR Health Technology Assessment Programme (project number 12/189/06). The views and opinions expressed in this paper are those of the authors and do not necessarily reflect those of the Health Technology Assessment Programme, NIHR, NHS or the Department of Health.

**Competing interests** None declared.

**Ethics approval** NRES Committee South West – Exeter.

**Provenance and peer review** Not commissioned; externally peer reviewed.

**Data sharing statement** The datasets analysed during the current study are not publicly available, as participants were not asked to consent to this at the time of data collection. However, if requests for data sharing are made, the University of Bristol Data Access Committee will consider them and decide whether or not they can be met.

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
