## [Reviewer comments · BMJ Open]

ARTICLE DETAILS

TITLE (PROVISIONAL)	Patients and nurses' views on providing psychological support within cardiac rehabilitation programmes: a qualitative study
AUTHORS	Turner, Katrina; Winder, Rachel; Campbell, John; Richards, David; Gandhi, Manish; Dickens, Chris; Richards, Suzanne

VERSION 1 - REVIEW

REVIEWER	Professor Alun C Jackson, Director Australian Centre for Heart Health, Melbourne, Australia
REVIEW RETURNED	16-Jun-2017

GENERAL COMMENTS	This is a timely and useful study on patients' and nurses' views on providing psychological support within cardiac rehabilitation services. I have a couple of questions and a more substantive comment. Page 4, lines 10-12. I would like to see some reference here not just to depression being associated with subsequent cardiac morbidity and mortality, but to some of the mechanisms of this relationship such as poor medication adherence, lack of exercise and poor diet. See for example: Murphy BM, Worcester MU, Goble AJ, Mitchell F, Navaratnam H, Higgins RO, Elliott PC, Le Grande MR. Lifestyle and physiological risk factor profiles six weeks after an acute cardiac event: are patients achieving recommended targets for secondary prevention? Heart Lung Circ 2011;20:446-451. Page 8 line 12 notes that "most" interviewees had attended a structured cardiac rehabilitation (CR) program. How many of the 18 interviewed patients was this? If not all attended, how could the EPC be offered if it was administered during CR? Page 8 line 41 "...interviewed at the time of interview" . Add "of" In the nurse interviews or in supervision, was any attempt made to judge the treatment fidelity. Do we know the extent to which the nurses followed the EPC guideline? If not then this should be acknowledged as a limitation of the study. Page 12 lines 3 to 21 note that UC patients had not discussed mental wellbeing. Did any of the interviewees say that they wanted to? If so, how many? I understand that this was a study of nurse provision of mental health support within CR, but I would be interested to see a brief discussion of what the research team believes is an alternative if nurses face barriers to implementation of a mental health
--

	intervention eg screening and referral to a mental health specialist. My more substantive comment relates to the conceptualising of the target for intervention through behavioural activation (BA), that constituted the enhanced psychological care intervention. In the introductory section BA is noted as "a simple psychological treatment for depression". There is a degree of slippage in terminology in the paper with reference variously to 'depression', 'depressive symptoms' and by the end of the paper, 'low mood'. This distinction is important, and relates to severity and appropriate action. I believe the authors are correct to talk of the intervention being for low mood, which almost all CR patients will experience and which has been identified as the "cardiac blues", and not as an intervention for depression, which is qualitatively different. I would like to see some discussion of this. For this distinction, see for example: Higgins RO, Murphy BM, Nicholas A, Worcester MU, Lindner H. Emotional and adjustment issues faced by cardiac patients seen in clinical practice: a qualitative survey of experienced clinicians. J Cardiopulm Rehabil Prev 2007;27:291-7. Murphy, B.M., Higgins, R.O., Jackson, A.C. Ludeman, D, Humphreys, J., Edington, J., Jackson, A., Worcester, M. (2015). Patients want to know about the 'cardiac blues'. Australian Family Physician 44(11), 826-32. Murphy, B.M., Higgins, R.O., Jackson, A.C. (2016). Anxiety, depression and psychological adjustment after an acute cardiac event, Handbook of Psychocardiology, (Eds. D. Byrne & M. Alvarenga), Springer. DOI:10.1007/978-981-4560-53-5_57-1
--	---

REVIEWER	Carl Lavie Ochsner Health System USA I am an author on many cardiac rehabilitation and psychological papers.
REVIEW RETURNED	23-Jun-2017

GENERAL COMMENTS	This is a very nicely written paper on an important topic. The major limitation is the very small N and lack of hard end-points, which would reduce its priority considerably for very competitive journals. However, this is worthy of publication and Open Access model. My colleagues have published a few recent papers on psychological aspects in cardiac rehabilitation patients which the authors could consider including: Lavie CJ et al Canadian Journal of Cardiology 2016;32:S365-S373; Sergey S et al American Journal of Medicine 2016;129:1316-1321; and Ernstsen L et al American Journal of Medicine 2016;129:82-88.
---

VERSION 1 – AUTHOR RESPONSE

Reviewer 1: Professor Alun C Jackson

This is a timely and useful study on patients' and nurses' views on providing psychological support within cardiac rehabilitation services. I have a couple of questions and a more substantive comment.

Many thanks for this positive feedback and for your helpful comments below.

Page 4, lines 10-12. I would like to see some reference here not just to depression being associated with subsequent cardiac morbidity and mortality, but to some of the mechanisms of this relationship such as poor medication adherence, lack of exercise and poor diet. See for example:

Murphy BM, Worcester MU, Goble AJ, Mitchell F, Navaratnam H, Higgins RO, Elliott PC, Le Grande MR. Lifestyle and physiological risk factor profiles six weeks after an acute cardiac event: are patients achieving recommended targets for secondary prevention? *Heart Lung Circ* 2011;20:446-451.

We have added the following text to the first paragraph in the Background section of the paper, and inserted the two new references into the reference list: Reasons for this association remain unclear but possible mechanisms include the association between depression and cardiac risk factors (e.g. hypertension, smoking, reduced physical activity), greater coronary disease severity, treatment non-adherence to cardiac medication and rehabilitation programmes, and increased platelet aggregation.^{8,9}

8. Carney RM, Freedland KE, Miller GE, Jaffe AS. Depression as a risk factor for cardiac mortality and morbidity. A review of potential mechanisms. *J Psychosom Res* 2002;897-902.

9. Dickens C. Depression in people with coronary heart disease: prognostic significance and mechanisms. *Curr Cardiol Rep* 2015;17:83.

We thank the reviewer for the Murphy et al (2011) reference. We read the paper with interest but decided to use two other references as they specifically focused on possible mechanisms between depression and subsequent cardiac morbidity and mortality.

Page 8 line 12 notes that "most" interviewees had attended a structured cardiac rehabilitation (CR) program. How many of the 18 interviewed patients was this? If not all attended, how could the EPC be offered if it was administered during CR?

Thank you for this comment. What we had written was not clear and we had confused attending the CR programme with attending fitness sessions, which were run as part of the programme. The text now reads: All interviewees had started their cardiac rehabilitation programme and attended at least one cardiac nurse appointment.

Page 8 line 41 "...interviewed at the time of interview" . Add "of"

We have now inserted the word 'of', so the text now reads 'nurses provided written consent to be interviewed at the time of interview'.

In the nurse interviews or in supervision, was any attempt made to judge the treatment fidelity. Do we know the extent to which the nurses followed the EPC guideline? If not then this should be acknowledged as a limitation of the study.

We have inserted the following text on page 6 (tracked version of the paper). In order to assess treatment fidelity, nurses were asked to complete a structured form after delivering EPC to patients. This form had been developed by the research team and invited nurses to record which elements of the EPC intervention they had delivered. These notes were reviewed by members of the research team, alongside feedback from the clinical supervision sessions. Based on this information, the research team concluded that the nurses had delivered EPC as intended.

Page 12 lines 3 to 21 note that UC patients had not discussed mental wellbeing. Did any of the

interviewees say that they wanted to? If so, how many?

We have inserted the following text on page 12 (tracked version of the paper):

Three of the UC patients mentioned they would have liked to have spoken to their nurse about their mental wellbeing, but then implied they had not been given the opportunity to do so.

'They [the nurses] are doing a brilliant thing, I cannot fault them... but they're very structured, they do what they do to help you and it's a great thing... it's just when you have got anxieties or you're worried about something... it's a shame, I felt that there wasn't enough of the one to one.' UC, patient 2

I understand that this was a study of nurse provision of mental health support within CR, but I would be interested to see a brief discussion of what the research team believes is an alternative if nurses face barriers to implementation of a mental health intervention e.g. screening and referral to a mental health specialist.

Thank you for this comment. We have already stated in the Conclusions and implications section of the paper that 'Given that nurses viewed provision within rehabilitation programmes as ideal, and mentioned good relations with local IAPT services, an alternative approach could be nurses co-ordinating IAPT-trained psychological wellbeing practitioners to provide psychological support within cardiac programmes.'

My more substantive comment relates to the conceptualising of the target for intervention through behavioural activation (BA), that constituted the enhanced psychological care intervention. In the introductory section BA is noted as "a simple psychological treatment for depression". There is a degree of slippage in terminology in the paper with reference variously to 'depression', 'depressive symptoms' and by the end of the paper, 'low mood'. This distinction is important, and relates to severity and appropriate action. I believe the authors are correct to talk of the intervention being for low mood, which almost all CR patients will experience and which has been identified as the "cardiac blues", and not as an intervention for depression, which is qualitatively different. I would like to see some discussion of this. For this distinction, see for example:

Higgins RO, Murphy BM, Nicholas A, Worcester MU, Lindner H. Emotional and adjustment issues faced by cardiac patients seen in clinical practice: a qualitative survey of experienced clinicians. *J Cardiopulm Rehabil Prev* 2007;27:291-7.

Murphy, B.M., Higgins, R.O., Jackson, A.C. Ludeman, D, Humphreys, J., Edington, J., Jackson, A., Worcester, M. (2015). Patients want to know about the 'cardiac blues'. *Australian Family Physician* 44(11), 826-32.

Murphy, B.M., Higgins, R.O., Jackson, A.C. (2016). Anxiety, depression and psychological adjustment after an acute cardiac event, *Handbook of Psychocardiology*, (Eds. D. Byrne & M. Alvarenga), Springer. DOI:10.1007/978-981-4560-53-5_57-1.

Thank you for identifying this inconsistency in terminology. From a conceptual perspective, our intervention was designed for people with moderate/severe depressive symptoms using cardiac rehabilitation services, i.e. not just recruiting people with the 'cardiac blues'. As explained in the methods section of the paper, individuals needed to score 10 or more on the PHQ-9 in order to be eligible to take part in the trial. While strictly speaking this score reflects depressive symptoms, as opposed to a clinical diagnosis of depression using a structured clinical interview, in reality, UK front-line primary care mental health services (e.g. GPs, IAPT mental health workers) routinely use this score to diagnose and actively manage/treat depression. In addition, whilst the PHQ-9 is not a diagnostic interview, it is very closely related to, and correlated with, the gold standard SCID diagnostic interview (Gilbody et al 2007, *BJGP* 2007;57:650-652).

Although the cardiac nurses screened for depressive symptoms, after early patient feedback on intervention design and study methods it became clear that a number of eligible patients did not identify with the terms 'depression' or 'depressive symptoms' despite sometimes scoring with severe symptoms on the PHQ-9. The term 'low mood' appeared more acceptable to both patients and nurses and thus we elected to use this term in patient facing materials, as it was more likely to engage a fuller spectrum of patients.

As clearly this switching of terminology is confusing, we have now re-read the paper and described the individuals recruited to the study as having depressive symptoms (acknowledging the fact that we did not give them a clinical diagnosis of depression) and removed the term 'low mood' and replace it with 'symptoms of depression' or 'how they (patients) felt'. In addition, we have added the following sentence to the Discussion section of the paper (under study strengths and weaknesses) to indicate that, although a clinical diagnosis of depression was not given, in the UK a score of 10 or more on the PHQ-9 is often used in this way, and to highlight that the PHQ-9 closely correlates with a diagnostic tool:

Individuals recruited to the CADENCE pilot trial needed to score 10 or more on the PHQ-9 in order to be eligible to take part. Strictly speaking this score reflects depressive symptoms, rather than a formal clinical diagnosis of depression. However, in the UK, front-line primary care mental health services (e.g. GPs, IAPT mental health workers) routinely use the PHQ-9 to diagnose and actively manage/treat depression. In addition, the PHQ-9 has been found to be as good as a diagnostic gold standard in detecting depression.³³

Reviewer 2: Carl Lavie

This is a very nicely written paper on an important topic. The major limitation is the very small N and lack of hard end-points, which would reduce its priority considerably for very competitive journals. However, this is worthy of publication and Open Access model. My colleagues have published a few recent papers on psychological aspects in cardiac rehabilitation patients which the authors could consider including: Lavie CJ et al Canadian Journal of Cardiology 2016;32:S365-S373; Sergey S et al American Journal of Medicine 2016;129:1316-1321; and Ernstsens L et al American Journal of Medicine 2016;129:82-88.

Many thanks for your comments and for the references. We have now cited and referenced the article by Sergey Kachur et al 2016 by inserting the following text in the Discussion section of the paper: In addition to suggesting that psychological support commenced earlier, patients also suggested it continued after discharge from a rehabilitation programme where necessary. Although evidence suggests depression after cardiac rehabilitation is not common, when present it is usually associated with other forms of psychological stress.²⁷ As there is an association between psychological stress and post-cardiac rehabilitation mortality, it has been suggested that patients with CHD are assessed for psychological risk factors both prior to and after receiving cardiac rehabilitation.²⁷

We read the other two articles with interest (thank you for drawing them to our attention) but felt they were less directly relevant to our paper.

VERSION 2 – REVIEW

REVIEWER	Professor Alun C Jackson Australian Centre for Heart Health, Australia
REVIEW RETURNED	12-Jul-2017

GENERAL COMMENTS	Thank you for considering the comments. This is important work and I look forward to seeing it in publication
---